# Comparative Analysis of Aristolochic Acids in *Aristolochia* Medicinal Herbs and Evaluation of Their Toxicities

**DOI:** 10.3390/toxins14120879

**Published:** 2022-12-16

**Authors:** Shu-Han Zhang, Yun Wang, Jing Yang, Dan-Dan Zhang, Yan-Lei Wang, Shu-Hui Li, Ying-Ni Pan, Hua-Min Zhang, Yi Sun

**Affiliations:** 1Institute of Chinese Materia Medica, China Academy of Chinese Medical Sciences, Beijing 100700, China; 2School of Traditional Chinese Materia Medica, Shengyang Pharmaceutical University, Shenyang 110016, China; 3Experimental Research Center, China Academy of Chinese Medical Sciences, Beijing 100700, China

**Keywords:** Aristolochiaceae, aristolochic acid, aristolactam, ultra performance liquid chromatography-quadrupole time-of-flight tandem mass spectrometry (UPLC-QTOF-MS/MS), orthogonal partial least squares-discriminant analysis (OPLS-DA), cytotoxicity, genotoxicity

## Abstract

Aristolochic acids (AAs) are a group of nitrophenanthrene carboxylic acids present in many medicinal herbs of the *Aristolochia* genus that may cause irreversible hepatotoxicity, nephrotoxicity, genotoxicity and carcinogenicity. However, the specific profile of AAs and their toxicity in *Aristolochia* plants, except for AAs Ι and ΙΙ, still remain unclear. In this study, a total of 52 batches of three medicinal herbs belonging to the Aristolochia family were analyzed for their AA composition profiles and AA contents using the UPLC-QTOF-MS/MS approach. The studied herbs were *A. mollissima* Hance (AMH), *A. debilis Sieb*.etZucc (ADS), and *A. cinnabaria* C.Y.Cheng (ACY). Chemometrics methods, including PCA and OPLS-DA, were used for the evaluation of the *Aristolochia* medicinal herbs. Additionally, cytotoxicity and genotoxicity of the selected AAs and the extracts of AMH and ADS were evaluated in a HepG2 cell line using the MTT method and a Comet assay, respectively. A total of 44 AAs, including 23 aristolochic acids and 21 aristolactams (ALs), were detected in *A. mollissima*. Moreover, 41 AAs (23 AAs and 18 ALs) were identified from *A. debilis Sieb*, and 45 AAs (29 AAs and 16 ALs) were identified in *A. cinnabaria.* Chemometrics results showed that 16, 19, and 22 AAs identified in AMH, ADS, and ACY, respectively, had statistical significance for distinguishing the three medicinal herbs of different origins. In the cytotoxicity assay, compounds AL-BΙΙ, AAΙ and the extract of AMH exhibited significant cytotoxicities against the HepG2 cell line with the IC_50_ values of 0.2, 9.7 and 50.2 μM, respectively. The results of the Comet assay showed that AAΙ caused relatively higher damage to cellular DNA (TDNA 40–95%) at 50 μM, while AAΙΙ, AMH and ADS extracts (ranged from 10 to 131 μM) caused relatively lower damage to cellular DNA (TDNA 5–20%).

## 1. Introduction

Aristolochic acids (AAs) are a group of nitrophenanthrene carboxylic acids mainly produced by plants of the *Aristolochia* and *Asarum* genera in the Aristolochiaceae family [1,2,3]. Currently, over 180 AAs analogues have been discovered, and AAΙ, AAΙΙ, AAΙΙΙa (AA C) and AAΙVa (AA D) are the most common in the *Aristolochia* genus [4]. It is worth noting that AAs are naturally occurring toxic compounds, and they could cause irreversible aristolochic acid nephropathy (AAN), as well as known toxicities including hepatotoxicity, nephrotoxicity, genotoxicity and carcinogenicity [5,6,7,8,9]. Toxicological studies have shown that metabolites of AAs could form AA–DNA adducts with DNA in target organs, and subsequently induce characteristic mutations of A-T transversion, which may be the reason of carcinogenesis [10]. Although AAΙ is considered to be the most toxic component to produce hepatotoxicity and nephrotoxicity, both AAΙ and AAΙΙ have been reported to be responsible for genotoxicity [4,11,12]. Due to the risk of human exposure to these toxic compounds, AAs have attracted increasing clinical attention, and have become a research hotspot worldwide [13,14,15].

Considering these noxious toxicities, the International Agency for Research on Cancer classified AAs as Group I carcinogens [16]. In addition, the Chinese Pharmacopoeia has continuously excluded four AA-containing herbal medicines of the *Aristolochia* genus since 2003 [17]. Although AA-containing herbs with abundant varieties, e.g., *A. debilis*, *A. contorta*, and *A. fangchi*, find limited use in China, many herbal materials and their derived prescriptions that contain AAs are still available on the market, and are in use due to clinical needs [18]. In 2017, the China Food and Drug Administration (CDFA) announced 24 herbal species, and more than 40 Chinese patent medicines derived from the *Aristolochia* genus are still in clinical use. Some AA analogues with lower content, such as aristolactam BΙ, BΙΙ and aristolochic acid ΙΙΙ a, have demonstrated their genotoxicity and cytotoxicity [4]. Therefore, further investigation on the use of AA limits is necessary.

As a result, the objectives of our study were to analyze AA compositions and their contents (Figure 1) of three representative herbal medicines from *Aristolochia* genera, *A. mollissima* Hance (AMH), *A. debilis Sieb*. et Zucc (ADS) and *A. cinnabaria* C.Y.Cheng (ACY), and to evaluate their toxicities against hepatic tumor cells. *A. mollissima* Hance (called Xun-Gu-Feng in China) is a common anti-rheumatic and analgesia medicinal herb in China [19,20,21], and mainly contains aristolochic acids Ι (AAΙ), ΙΙ (AAΙΙ), ΙΙΙa (AA C) and aristolactam Ι (AL–Ι) [22,23]. *A. debilis Sieb*.etZucc (Tian-Xian-Teng) is used in the form of the aerial part of the plant. It can be used to relieve abdominal pain and rheumatic arthralgia [24]. *A. cinnabaria* C.Y.Cheng (Zhu-Sha-Lian) is the root of *A*. *cinnabaria* C.Y.Chengtj.L.wu, and has positive effects for the treatment of enteritis, dysentery and sore throat, etc. [25,26,27].

Although the main toxic aristolochic acids Ι and ΙΙ have been found in the Aristolochia genus, the toxicity of their analogues and their specific presence in the Aristolochia plant are uncertain. Importantly, medicinal herbs containing AAs are still commonly used in certain Chinese patent drugs, such as Shedan Chuanbei powder, Duzhong Zhuanggu capsule and Hewei jiangni capsule, etc. Therefore, it is necessary to conduct a thorough study on the existence of AAs in Aristolochia herbs, in order to reasonably control AAs in such medicinal herbs and related Chinese patent drugs.

**Figure 1 toxins-14-00879-f001:**
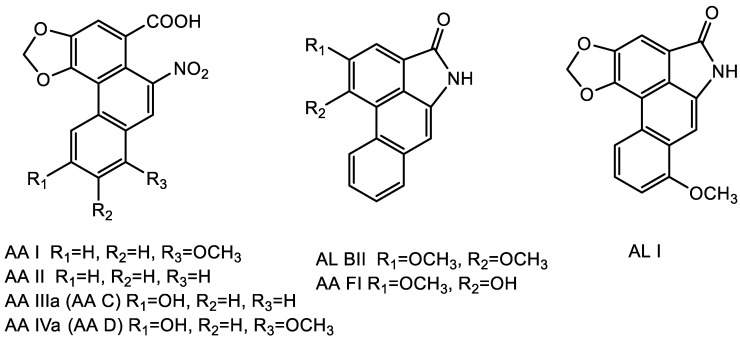
Chemical structures of AAs in the Aristolochiacea family.

## 2. Results and Discussion

### 2.1. Identification of AAs Components by UPLC-QTOF-MS/MS

The UPLC-QTOF-MS/MS technique was applied to identify AAs in the three species of the *Aristochia* genus herbs, including AMH, ADS and ACY. The chemical structure of each individual AA was identified based on the chemical standard and their chromatographic and mass spectrometric properties, such as elementary composition analysis, the retention behavior and mass fragments reported in the literature [13,27,28]. The MS data libraries were also referred to using the TCM Systematic Pharmacology Database and Analysis Platform (TCMSP), the GNPS database and Waters Traditional Medicine Library, The University of Mississippi Botanical Library, and the University of Ottawa Phytochemical Library.

The high resolution of TOF-MS provided accurate mass determination and calculated reasonable chemical formulas according to molecular weight. AA structures were classified into two groups, aristolochic acids and aristolactams, as identified by their chemical formulas and typical mass fragments (Figure 2 and Appendix A). A total of 76 compounds, including 44 AAs (Table 1 and Figure 3) and 32 other types of components (Appendix A), were detected in AMH. In addition, 72 compounds in total were determined from ADS, including 41 AAs (Table 1 and Figure 3) and 31 other compounds (Appendix A), and 45 AAs were identified in *A. cinnabaria* (Appendix A).

MS^n^ data of AAΙ at *m*/*z* 342.0623 [M + H]^+^ yielded a series of characteristic ions at *m*/*z* 324.0530 [M + H − H_2_O]^+^, 298.0743 [M + H − CO_2_]^+^ and 296.0707 [M + H − NO_2_]^+^, resulting from the neutral losses of H_2_O, carbon dioxide and nitrogen dioxide, respectively (Figure 2). Basically, the neutral losses of H_2_O, CH_3_, H_2_OCH_3_, CO, CO_2_ and NO_2_ are the characteristic fragments of aristolochic acids with a nitrophenanthrene skeleton. For example, AAIV exhibited an [M + H]^+^ ion at *m*/*z* 372.0709, and the MS^2^ spectrum showed key characteristic ions at *m*/*z* 354.0637, 326.0825, 311.0778 and 283.0682, which were mainly attributed to losses of H_2_O, NO_2_, NO_2_CH_3_ and NO_2_CH_3_CO, respectively.

Aristolactams were also analyzed using MS^n^ determination, and exhibited similar neutral losses to those of aristolochic acids. The key fragments of AL-Ι, AL-AΙΙ and AL-Ιa-β-D-glucoside are shown in Figure 2. AL-Ι exhibited an [M + H]^+^ ion at *m*/*z* 294.0774, and the MS^2^ spectrum showed several key characteristic ions at *m*/*z* 279.0579, 251.0633 and 221.0560, which were mainly attributed to losses of CH_3_, CH_3_CO and CH_3_COCH_2_O, respectively. Aristolactam AΙΙ exhibited an [M + H]^+^ ion at *m*/*z* 266.0812. Moreover, the MS^2^ spectrum displayed several key characteristic ions at *m*/*z* 251.0534, 223.0597, 195.0631 and 167.0700, which were mainly attributed to losses of CH_3_, CH_3_CO, CH_3_COCO. and CH_3_COCOH_2_O, respectively.

### 2.2. Multivariate Statistical Analysis

#### 2.2.1. Semi-Quantitative Analysis of AAs

As shown in Figure 4, Figure 5 and Figure 6, and in Table 1 and Appendix A, we firstly compared the contents of AAs and ALs between ACY and the other two medicinal herbs. We found that the contents of 26 AAs and ALs in ACY were higher than those in ADS and AMH. For instance, compounds such as cinnabarin, AL-IVa-O-β-D-glucoside, aristoliukine C, aristophyllides C, aristchamic A, 7-OH-AAΙ and AAΙ, a methyl ester, were only detected in ACY. Moreover, the primary components of AAΙ, AAΙΙ, AA C and AL-Ι produced by ACY exceeded by about 30–400% those in AMH, but AAD and AL-FΙ in ACY were 50% less than those in AMH. Secondly, compounds 9-OH-AAΙ, AA E, ariskanin E, and AL-BΙΙ were determined to exist only in AMH. However, compounds Ariskanin B and aristoloterpenate IV were only detected in ADS. AAΙ and AAΙΙ in ADS were reduced by 30–60% compared with AMH, but its AA D and AL-Ι were 200–400% less than those in AMH. It can be speculated that the toxicity caused by ACY was the most potent.

The medicinal herb AMH collected in Hubei and Shandong Provinces exhibited great differences in AA content. Compounds AL-ΙΙΙa-N-β-D-glucoside, AL-Ι-N-β-D-glucoside, AL-AΙΙΙa, AL-ΙΙ, AL-ΙΙ-N-β-D-glucoside and AL-Ιa-N-β-D-glucoside in AMH, which originated from Hubei province, increased by 800–1200% compared to the ALs from herbs from Shandong province. Similarly, compounds AL-Ι-N-β-D-glucoside, aristoliukine A, AL-ΙΙΙa-N-β-D-glucoside, aristoloterpenate ΙV, 6-methoxydenitroaristolochic acid methyl ester, and aristolic acid ΙΙ-8-O-β-D-glucoside in ADS from Guangxi province decreased by 5–30% compared to ADS from Hubei province. Additionally, compounds aristchamic B, AL-Ι-N-β-D-glucoside, aristophyllides C, AL-ΙΙ and AL-Ιa in ACY from Sichuan province showed a 200–300% increase compared with those from Yunnan province. In this case, the possibility of kidney and liver injury may increase, if the patients take these toxic components in the long term.

#### 2.2.2. Results of PCA and OPLS-DA

Unsupervised principal component analysis (PCA) was established using AA contents as the variables in order to distinguish herbal samples collected from different species and origins (Figure 4). All the mass spectroscopic data acquired by UPLC-Q-TOF-MS/MS were converted into a three-dimensional matrix, including the retention time, *m*/*z* values, and peak intensities. A total of 66 variables were generated and subjected to PCA analysis on the SIMCA software. The PCA scores plot showed a considerable separation tendency among the three species of AMH, ADS and ACY, with an R^2^X of 0.609. Distinct from other samples, all the samples of ACY cluster were in one region. Samples of ADS were in the lower left quadrant, while the samples of AMH were in both the upper and lower left quadrants (Figure 4A). As shown in Figure 4B, the PCA scores plot that displayed the separation between Hubei and Shandong provinces were obviously significant (R^2^X = 79.7%), indicating that the AAs/ALs contents of the AMH from the two provinces were different. However, the PCA scores plot of ADS (R^2^X = 0.634) and ACY (R^2^X = 0.704) could not reveal a considerable separation either between Hubei and Guangxi provinces, or Sichuan and Yunnan provinces, indicating that the samples of ADS and ACY from the above four provinces could not be well distinguished according to the content of the AAs/ALs (Figure 4C,D).

**Figure 4 toxins-14-00879-f004:**
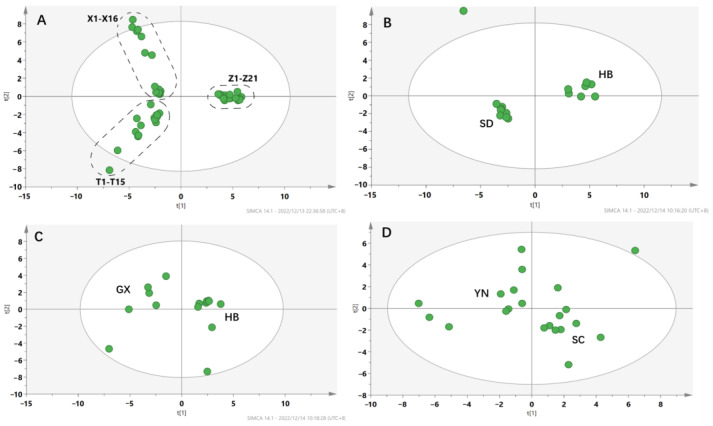
PCA score scatter plot for herbal samples collected from different species and origins. (**A**) Three species of medicinal samples, including batches X1−X16 of AMH, batches T1−T15 of ADS, and batches ZI−Z21 of ACY; (**B**) AMH (HB: Hubei; SD: Shandong); (**C**) ADS (HB: Hubei; GX: Guangxi); (**D**) ACY (SC: Sichuan; YN: Yunnan).

To evaluate the intra-group differences among the samples from different species and origins, supervised OPLS-DA was further performed to obtain the corresponding model. As shown in Figure 5A, all the samples were obviously clustered into the distinct groups corresponding to the species (R^2^X = 0.627, R^2^Y = 0.982, Q^2^ = 0.922).

**Figure 5 toxins-14-00879-f005:**
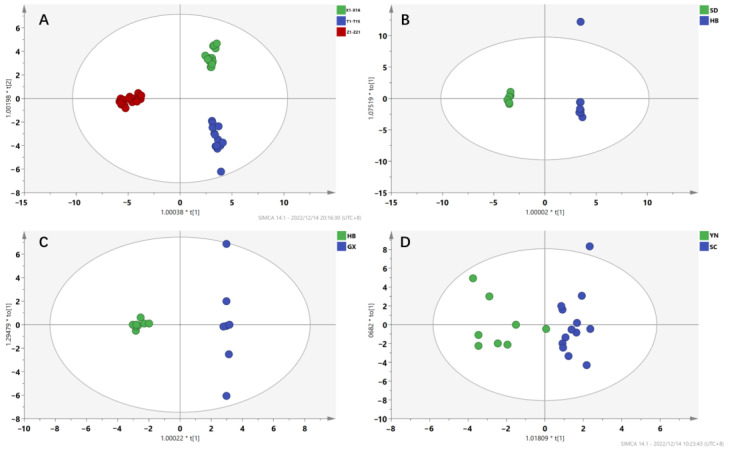
OPLS-DA score scatter plot for three medicinal samples collected from different origins. (**A**) Three species of medicinal samples, including batches X1−X16 of AMH, batches T1−T15 of ADS, and batches ZI−Z21 of ACY. (**B**) AMH (HB: Hubei; SD: Shandong); (**C**) ADS (HB: Hubei; GX: Guangxi); (**D**) ACY (SC: Sichuan; YN: Yunnan).

In Figure 5B, OPLS-DA analysis gave similar robust differentiation of the AMH samples from Hubei and Shandong provinces (R^2^X = 0.764, R^2^Y =0.995, Q^2^ = 0.968). The OPLS-DA scores plot also showed an obvious separation between Hubei and Guangxi provinces, with R^2^X of 0.740, R^2^Y of 0.995, Q^2^ = 0.803 (Figure 5C), indicating that the AAs/ALs contents of ADS from the two provinces were different. However, the OPLS-DA scores plot of ACY could not reveal an obvious separation between Sichuan and Yunnan provinces, indicating that the samples of ACY from the two provinces could not be well distinguished according to the contents of the AAs/ALs (Figure 5D).

#### 2.2.3. Identification of Differential Components

Compounds with variable importance in projection (VIP) values larger than 1 and *p* < 0.05 were viewed as differential compounds and discriminate quality markers. On the basis of the results of VIP value and t test, the AAs labeled with different origins were screened and identified.

A total of 33 significant markers were determined to facilitate discrimination of three kinds of medicinal materials (Figure 6A). The components were tentatively identified as cinnabarin, aristoliukine C, 9-OH-AAI, AAI, 7-OH-AAI, AL-I, AL-Ia-N-β-D-glucoside, aristoloterpenate IV, ariskanin E, AL-BIII, AAII, AL-IVa-O-β-D-glucoside, AAD, AL-FI, AAE, Ariskanin-B, AL-II-N-β-D-glucoside, AL-AIIIa, 10-amino-5,7-dimethoxy-aristolic II, AAIa methyl ester, AL-I-N-β-D-glucoside, AL-CV, AAC, AL-III, aristchamic A, aristophyllides C, Aristololactamoside II, aristolic acid I, AAIII, AL-IIIa-N-β-D-glucoside, AL-IVa, 6-methoxydenitroaristolochic acid methyl ester and AAI methyl ester.

**Figure 6 toxins-14-00879-f006:**
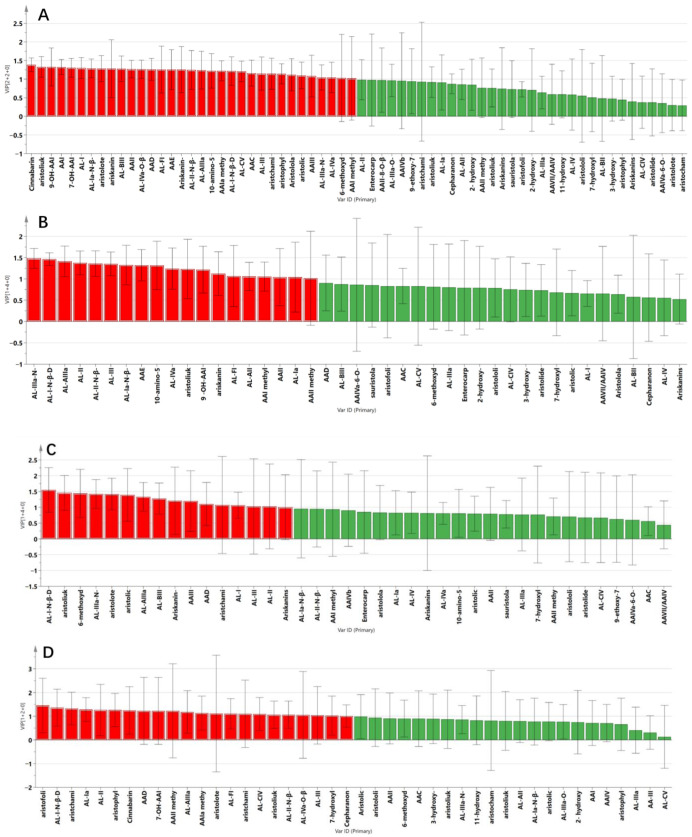
Important variables analysis for three medicinal samples collected from different origins. AAs with variable importance in projection (VIP) scores greater than 1 were considered important compounds towards the classification model and are marked in red. Compounds with VIP < 1 are marked in green. (**A**) Three medicinal materials of AMH, ADS and ACY; (**B**) AMH (HB: Hubei; SD: Shandong); (**C**) ADS (HB: Hubei; GX: Guangxiy); (**D**) ACY (SC: Sichuan; YN: Yunnan).

A total of 19 differential markers from AMH samples in Hubei and Shandong province,16 differential markers from ADS samples in Hubei and Guangxi province and 22 differential markers from ACY samples in Sichuan and Yunnan province were screened out, respectively (Figure 6B−D). The above AA components may be the main reason for the differences in the three Aristolochiaceae medicinal herbs from different origins.

### 2.3. Quantification of AAs in Aristolochia Medicinal Herbs

#### 2.3.1. Method Validation

An efficient UPLC-QTOF-MS/MS method for quantitative analysis was validated based on linearity, limits of detection (LOD), limits of quantitation (LOQ), intra-day and inter-day precision, stability, and accuracy (Table 2 and Table 3). All calibration curves showed good linearity (R^2^ > 0.9964) within the test ranges (Table 3). This analysis method was experimentally demonstrated to have good precision, stability and reproducibility (RSD < 5%). The recovery of quantitative AAs ranged from 97.23% to 103.19% (RSD from 3.87% to 4.68%, respectively) (Table 2). In addition, the peak purity was investigated by analyzing the UPLC and MS/MS data, and no indication of impurity was found. The results indicated that the proposed method was effective and accurate for the determination of AA standards.

#### 2.3.2. Quantitative Analysis of AAs in Herbal Samples in Aristolochia

AAs were mainly distributed in the medicinal plants of Aristolochia family, such as *A. debilis*, *A. contorta*, *A. fangchi*, *A. mollissima* and *A. cinnabaria.* In this study, fifty-two herbal samples, belonging to three representative AA-containing species in the Aritolochia family, were analyzed for their content of AAs. The samples included 15 batches of aerial part of *A. debilis* (ADS), 16 batches of *A. mollissima* (AMS) herb, and 21 batches of *A. cinnabaria* (ACY) root. The AAs exhibited strong mass spectrum signals in the positive mode. Multiple reaction monitoring (MRM) of UPLC-QTOF-MS/MS was performed for quantification. Indometacin was used as an internal standard.

The established analysis method was suitable for the quantification of seven AAs in AMH sample and five AAs in the ADS sample (Figure 7). As can be seen from the results in Appendix A, AAs were detected in all herbal samples, with significant variations in total amounts. MRM chromatograms of the components and standards are shown in Figure 7. The contents of AAΙ, AAΙΙ, AA C, AA D, AL-Ι, AL-BΙΙ and AL-FΙ in 16 batches of AMH were in the ranges of 123.659–600.260, 38.388–198.685, 7.626–29.256, 2.550–70.857, 3.365–15.466, 0.022–0.235 and 1.274–22.537 μg/g, respectively (Appendix A), with the content of AAΙ being the highest. AAΙ and AAΙΙ were undoubtedly the main toxic components, with high contents in AMH. Moreover, the contents of AAΙ, AAΙΙ, AA C, AA D, and AL-Ι in 15 batches of ADS were 6.734–17.256, 4.438–33.322, 0.945–2.297, 2.792–120.623, and 0.894–32.102 μg/g, respectively (Appendix A). As a result, as shown in Appendix A, ADS contained a smaller amount of AAs compared to the other two herbs. However, AA D possessed the highest contents among the quantified AAs in ADS, a feature which was different from most medicinal herbs in the Aristolochia genus. The determination of the AA contents in ACY was part of a previous study by our team (Appendix A), where quantitative results were compared with the above AA contents in AMH and ADS [27].

The contents of medicinal herbs from different origins were compared, and it was found that the origins had obvious impact on the AA contents of the three medicinal herbs (Figure 8A–C). The three groups of herbal samples with high AA contents could be easily visualized according to their different origins. Furthermore, each AA’s content could be easily compared with the other analogues in all three herbs of the Aristolochia family (Figure 8D). The total contents of AAs in ACY were considerably higher than those in the other species of AMH and ADS. Thus, we could conclude that ACY is the most toxic.

**Figure 7 toxins-14-00879-f007:**
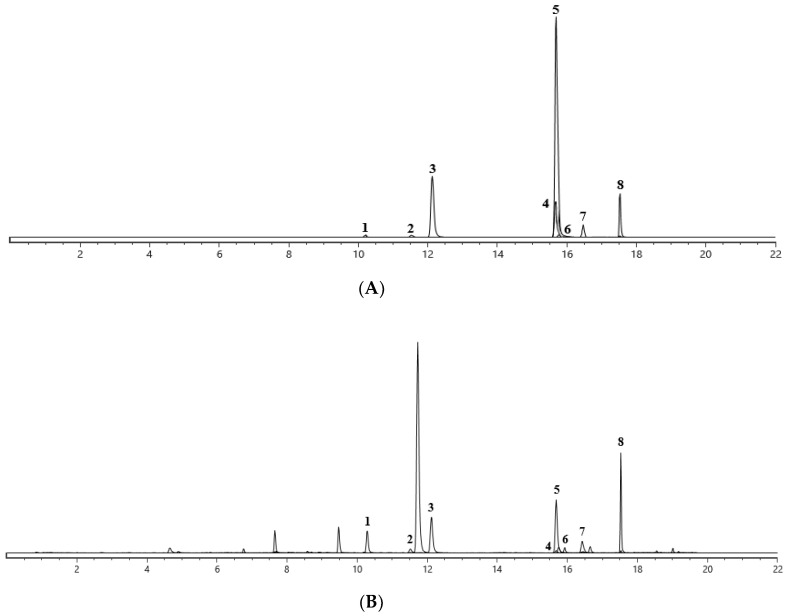
Multiple reaction monitoring (MRM) chromatogram of standard compounds and herbal extracts. (**A**) The reference substances; (**B**) AMH sample; (**C**) ADS sample. 1—AA C; 2—AA D; 3—AL-FⅠ; 4—AL-BII; 5—AL-I; 6—AAII; 7—AAI; 8—Indometacin.

### 2.4. Cytotoxicity and Genotoxicity of AAs and the Extracts

AAΙ and AAΙΙ are the most common aristolochic acids produced by the plants of the Aristolochiaceae family. Both nitro and methoxy groups in their structures exert the toxicity, and substitutions of nitro groups possess more toxicity those of methoxy. Similarly, aristolactams are divided into two types, Ι and ΙΙ, according to their structures, with or without the methylenedioxy group, respectively. Some of the type ΙΙ ALs displayed cytotoxicities against the tumor cells [9]. AAs and their analogues have been reported to implicate liver cancer, suggesting that their related herbal extracts and their products may induce hepatotoxicity [29]. Therefore, in this study, we evaluated cytotoxicity against HepG2 tumor cells and genotoxicity of AAs and the extracts of AMH and ADS.

The results of the cytotoxicity assay indicated that AAΙ showed moderate cytotoxicity against HepG2 cell line, but AAD had no cytotoxicity. Among the AA derivatives, AL-BΙΙ exhibited the most potent cytotoxicity, and possessed selective cytotoxicity against the NCI-H187 cell line, but no cytotoxicity against A549 and MCF7 [30,31]. Particularly worth mentioning is that AMH extract displayed potent toxicity with an IC_50_ value of 9.7 µM, but ADS extract only had weak cytotoxicity (IC_50_ value of 92.9 µM) against the HepG2 cell line (Table 4).

Comet assay was used to evaluate the effects of AAI, AAII, ADS and AMH extracts on DNA damage of the HepG2 cells (Figure 9 and Appendix A).

Statistical significance for the comet assays was determined by analysis of Kruskal–Wallis test with Bonferroni correction. In Figure 9A, Kruskal–Wallis test comparisons revealed significant differences in the tail DNA content between the blank control group and the other four groups, while the AAII group showed no significance (Kruskal–Wallis test comparisons of blank vs. AAΙ or AMH or ADS or ADM, *p* < 0.001; and blank vs. AAΙΙ, *p* > 0.05). The tail DNA content of AAI was much higher than those of AAΙΙ and ADS, increasing by 16 times and 4.7 times, respectively, and exhibited no obvious difference with that of the positive control ADM group (Dunn’s test comparisons of AAΙ vs. AAΙΙ or ADS, *p* < 0.001; and AAI vs. ADM, *p* > 0.05). The above results indicate that AAΙ obviously caused damage to the DNA. Therefore, AMH with high AAΙ causes more serious damage than ADS.

As shown in Figure 9B, the tail length of the AAΙ, AMH, ADS and ADM groups was significantly higher than that of the blank control group, while the tail DNA content of AAΙΙ displayed no significance (Kruskal–Wallis test comparisons of blank vs. AAΙ or AMH or ADS or ADM, *p* < 0.001; and blank vs. AAΙΙ, *p* > 0.05). Similarly, the tail length of AAI was much higher than that of AAΙΙ and ADS, increasing by 12.8 times and 6.8 times, respectively, with no significant difference from those of the positive control ADM group and the AMH group (Dunn’s test comparisons of AAI vs. AAII or ADS, *p* < 0.001; and AAI vs. ADM or AMH, *p* > 0.05).

The tail moment values of AAΙ, AMH, ADS, ADM, and AAΙΙ groups were significantly higher than that of the blank control group (Kruskal–Wallis test comparisons of blank vs. AAI or AMH or ADS or ADM or AAII, *p* < 0.001). Surprisingly, AAΙ and ADM exhibited markedly enhanced comet tail moment, reaching values significantly higher than other groups. Compared with AAΙΙ group, the value of tail moment of AAΙ significantly increased, by 110 times. Moreover, the value of tail moment of AAΙ increased approximately 70 times compared with ADS group. However, there were no obvious differences between the AAΙ group and the AMH and ADM groups (Dunn’s test comparisons of AAI vs. ADS, *p* < 0.001; and AAI vs. ADM or AMH, *p* > 0.05) (Figure 9C).

Combined with the cytotoxic results, the comet assay indicated that both medicinal herbs could cause cellular DNA damage. AMH was more toxic than ADS, which was consistent with the quantitative results. The AAΙΙ group was not statistically significant compared with the negative control group, indicating that AAΙΙ caused no obvious damage to the DNA of HepG2 cells.

## 3. Conclusions

In this study, UPLC-QTOF-MS/MS was used to establish a qualitative and quantitative method for the analysis of AA components of Aristolochia herbs. A comparative analysis of AAs was applied to different medicinal herbs in Aristolochia family. Over forty AA analogues, including AAs and ALs types, were identified from each studied herb, whereby AAΙ was the principle component in both AMH and ACY. However, AAΙVa (AA D) was the main component in ADS. It is worth mentioning that ACY possesses large amounts of AAsΙ, ΙΙ, ΙΙΙ and ΙVa, which should be the most toxic among the three medicinal herbs. Chemometric results indicate that more than 14 AAs have statistical significance for differentiating the three herbal samples from different origins. The toxicity of AAs and the herbal extracts were evaluated by MTT method and Comet assay in HepG2 cell line. The compound of AL-BΙΙ exhibited the most potent toxicity, and the herbal extract of AMH also had strong toxicity. Similarly, comet assay showed that the DNA damage caused by AMH was greater than that caused by ADS. Thus, the quantitative results are consistent with the experimental data on toxicity, suggesting that long-term exposure to high levels of AMH and ADS would cause a risk of liver injury. The dosages of AMH and ADS should be paid more attention and limited in the clinical application. Our findings will have important significance for the safety control of such AA-containing herbs and their related prescriptions.

## 4. Materials and Methods

### 4.1. Chemicals and Materials

UPLC-QTOF-MS/MS data were obtained on a Waters Vion QTOF/MS (Waters Micromass, Manchester, UK) in positive electrospray ionization mode, for which the software MassFragment TM 1.9.4.053 was used. An AUW120 electronic analytical balance (Shimadzu Company, Kyoto, Japan), a Dy6c electrophoresis instrument power supply (Beijing Liuyi Instrument Factory, Beijing, China), and a Nikon Eclipse 80i fluorescence microscope (Nippon Nikon Company, Tokyo, Japan) were used for the Comet assay.

Eight standard compounds were purchased from Chengdu Pusi Bio-Technology Co., Ltd. (Chengdu, China), including AAΙ (PS010645), AAΙΙ (PS010031), AA C (PS010029), AA D (PS010039), AL-Ι (PS010658), AL-BΙΙ (BBP01017), AL-FΙ (BBP01020) and indomethacin (9AEUSEHD). The purity of each compound was above 98%. LC-MS grade solvents of methanol and acetonitrile were purchased from Fisher Scientific, and deionized water was obtained from Watsons (Guangzhou, China). All other chemicals and reagents were of analytical grade and were purchased from Beijing Chemical Works (Beijing, China). DMEM medium, fetal bovine serum, and trypsin were purchased from cytiva Co., Ltd. (Guangzhou, China). Comet assay kit was purchased from ELK Biotechnology Co., Ltd. (Wuhan, China). Human hepatocellular carcinoma cells HepG2 were purchased from Procell Life Science & Technology Co., Ltd. (Wuhan, China).

AMH were purchased from Hubei and Shandong provinces, and ADS was purchased from Hubei and Guangxi provinces in China. Their botanical species of the medicinal samples were identified by Professor Pan Yingni. The medicinal samples were deposited in the China Academy of Traditional Chinese Medicine and the Chinese Medicine Research Institute.

### 4.2. Preparation of Standard and Sample Solutions

The stock solutions were prepared separately by dissolving the reference compounds in methanol to obtain solutions of AAΙ, AAΙΙ, AA C, AA D, AL-Ι, AL-BΙΙ, and AL-FΙ, respectively. Then, fix the volume in a volumetric flask and prepare the mixed reference substance solution with the corresponding mass concentrations of 140.00, 90.00, 20.00, 30.00, 10.00, 0.20, and 9.00 μg/mL, respectively, and store at 4 °C for later use. The stock solution was prepared by dissolving the internal standard substance in methanol to obtain a solution of indomethacin (50.00 μg/mL), which was stored at 4 °C for later use.

A total of 5.0 g powdered sample was extracted with 100 mL 75% ethanol for 2 h under reflux. The organic solvent was removed under reduced pressure to yield the extract. Then the extract was filled to a 25 mL volumetric flask with methanol to get the test solution. Then, 180 μL of the test solution and 20 μL of the internal standard solution were mixed and successively filtered through a 0.22 μm membrane filter before injection. Each sample preparation and injection were repeated. The solution was stored at 4 °C before quantitative analysis.

For the qualitative experiment, the above ethanol sample was further extracted with petroleum ether and ethyl acetate. The ethyl acetate layer was concentrated and dissolved in methanol to make a solution of 0.5 mg/mL. Finally, the solution was filtered through a 0.22 μm membrane filter before injection. The preparation and injection of each sample was repeated. The solution was stored at 4 °C.

### 4.3. UPLC-QTOF-MS Conditions

Ultraperformance liquid chromatography–tandem mass spectrometry was performed using Waters UHPLC system, coupled to a Q-TOF MS/MS spectrometer (Waters, Milford, CT, USA). The chromatographic separation was carried out on a Waters Acquity ACQUITY UPLC-BEH C18 column (2.1 mm × 100 mm, 1.7 μm) with the flow rate of 0.3 mL/min. The column was set at 35 °C, and the injection volume was 1 μL. The mobile phase was composed of 0.1% formic acid aqueous solution (A)–acetonitrile (B). The gradient elution conditions were set as follows: 0–2 min, 10% B; 2–7 min, 10–30% B; 7–12 min, 30–35% B; 12–15 min, 35–50% B; 15–18 min, 50–95% B; 18–22 min, 10% B.

For MS detection, a Q-TOF MS spectrometer was fit with electrospray ionization (ESI) in positive ionization mode at full scan mode ranged *m*/*z* 50–1500. The MS parameters were as follows: cone voltage 30 V, capillary voltage 3.0 kV, cone gas flow rate (N_2_) 50 L/h, desolvation gas flow rate 800 L/h, ion source temperature 120 °C, desolvation gas temperature 450 °C, and spectrum acquisition frequency 0.2 s.

All the data were analyzed using UNIFITM1.9.4.053, Progenesis QI, simca14.0 and GraphPad Prism 7. The mass spectrum parameters of quantitative compounds are shown in Table 5.

### 4.4. Method Validation for Quantification

#### 4.4.1. Calibration Curves, Limits of Detection and Quantification

Using the peak area Y (the ratio of the peak area of the reference to that of the internal standard) versus the mass concentration X (the ratio of the reference to the internal standard), the standard curve was drawn and calculated to obtain a linear regression equation and a linear range. The limits of detection (LODs) and limits of quantification (LOQs) of the samples were estimated at signal-to-noise ratios (S/N) of 3 and 10, respectively.

#### 4.4.2. Precision, Stability, Repeatability and Recovery

To verify the precision, Sample X03 was injected six times consecutively and the RSD values of peak areas were calculated. To verify the stability, Sample X03 was analyzed at 0, 2, 4, 8, 12, 16, 20, and 24 h, respectively. To verify the repeatability, we prepared six test samples and performed six replicate analyses. The variability is expressed as RSD (%). The recovery experiment was determined by the standard addition method. The controls of seven components equal to the content in sample X03 were added precisely. The recoveries were calculated based on the difference in mass of these standards before and after addition.

### 4.5. Multivariate Statistical Analysis and Comparison

The data files for the qualitative components and peak areas of 16 batches of AMH (from Hubei and Shandong), 16 batches of ADS (from Hubei and Guangxi) and 21 batches of ACY (from Sichuan and Yunnan) were exported using Progenesis QI software and then imported to simca14. 0 software for PCA principal component analysis. The supervised pattern analysis of OPLS-DA was performed according to the different origins of the same medicinal material and three different medicinal herbs. The compounds that contributed significantly to the isolation between groups were initially screened (VIP value > 1) according to variable importance in projection (VIP), and potential differentially labeled compounds were screened based on independent sample *t*-test (*p* < 0.05 for statistically significant difference).

### 4.6. Cytotoxicity Assay and Comet Assay

#### 4.6.1. Cytotoxicity Assay

The cytotoxicity of the compounds to HepG2 was determined using the MTT method. First, the cells were transferred from the culture dish to a centrifuge tube, and centrifuged for 5 min (at 1000 r/min). After discarding the supernatant, the tumor cells were cultured in a complete medium containing 15% fetal bovine serum at 37 C, 5% CO_2_, and saturated humidity. When the cells grew logarithmically, they were spread onto plates, and the cell density was adjusted to 8.0 × 10^4^ cells/mL before the cell sap was added into a 96-well plate (200 µL per well). The cells grew adherent for 24 h. After 24 h, 2 µL of two standards, two extracts of medicinal herbs were added to each plate well and incubated for 72 h. Adriamycin was used as the positive control, and the blank control contained 2 µL DMSO. After incubation, 20 μL of 5 mg/mL MTT solution was added, and the plates were incubated for 4 h. The supernatant liquid was removed, and the cells were disrupted with 200 µL of DMSO for 10 min. After the blue crystalline substance was fully dissolved, the absorbance value was measured on a microplate reader with the detection wavelength of 562 nm and the reference wavelength of 630 nm. The IC_50_ value of each compound was calculated.

#### 4.6.2. Comet Assay

Cells in the logarithmic growth phase were inoculated in 6-well plates, and the cell density was adjusted to 2.5 × 10^5^ cells /mL, 2 mL per well. After 24 h, the experimental group was combined with AAⅠ (50 μM), AAII (131.5 μM), extracts of the AMH (10 mg/mL) and ADS (93 mg/mL), the positive control group was combined with ADM (3.4 μM), and the blank control group were added with the same volume of cell culture solution. After 48 h of culture, the cells were collected. DNA damage was detected according to the operating instructions of comet assay kit. The experiment was repeated 3 times, and 50 cells were selected at each dose in each experiment and analyzed by Comet Assay software (CASP, http://casplab.com/, accessed on 5 October 2022). The mean values of tail DNA percentage, tail length, and tail moment of three experiments were used to express DNA damage. Results one-way ANOVA was used to test the data between groups, and the difference was statistically significant when *p* < 0.01. Data and statistical results are generated by GraphPad Prism 7 software.

## Figures and Tables

**Figure 2 toxins-14-00879-f002:**
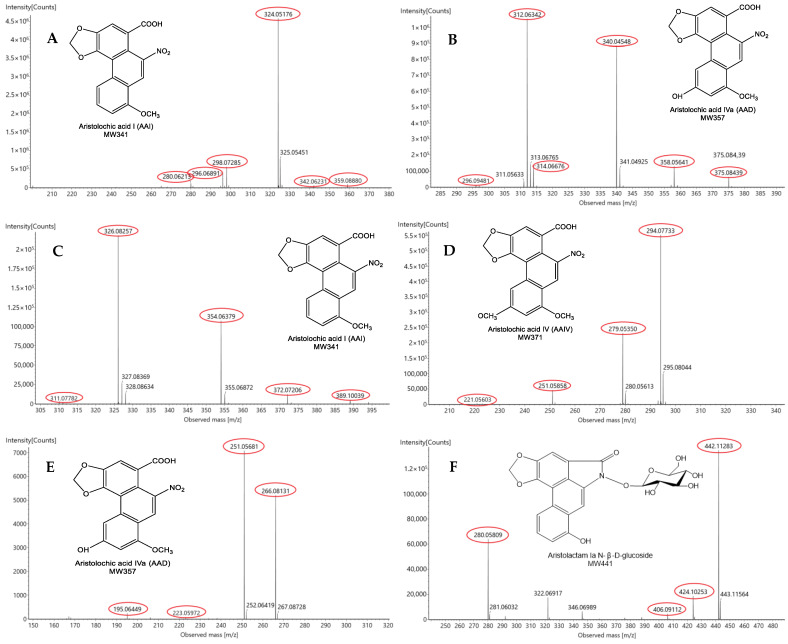
The MS^e^ spectrum of six AAs and ALs. (**A**) AA Ⅰ; (**B**) AA D; (**C**) AA Ⅳ; (**D**) AL-Ⅰ; (**E**) AL-AⅡ; (**F**) Aristolactam Ia N-β-D-glucoside. (Red circles indicate the MS fragments for each compound.)

**Figure 3 toxins-14-00879-f003:**
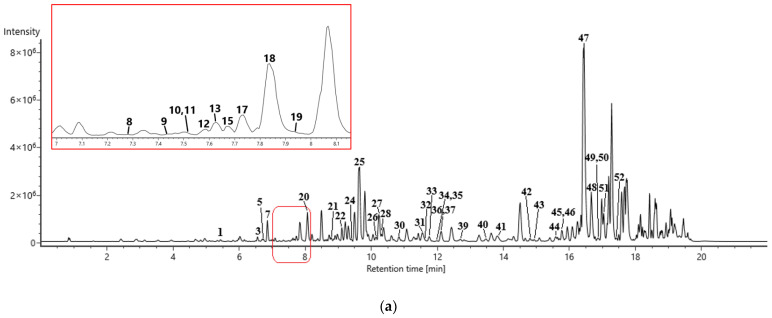
The total ion chromatograms (TIC) of the AMH extract (**a**) and ADS extract (**b**) under positive ion mode by UPLC-Q-TOF-MS/MS analysis.

**Figure 8 toxins-14-00879-f008:**
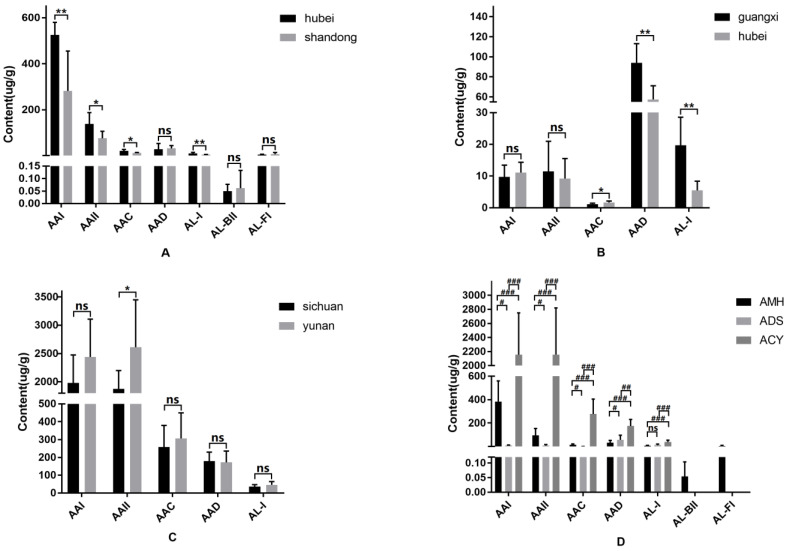
Content differences of the AAs in AMH, ADS and ACY from different origins (5A–5C), respectively; and content differences in 3 species of medicinal herbs (5D). (**A**) AMH of Hubei and Shandong provinces; (**B**) ADS of Guangxi and Hubei provinces; (**C**) ACY of Sichuan and Yunnan provinces; (**D**) three medicinal herbs AMH, ADS and ACY. Data are expressed as the mean ± SD. *: *p* < 0.05 and **: *p* < 0.01 analyzed by Mann Whitney U test. # represents *p* < 0.05; ## represents *p* < 0.01; ### represents *p* < 0.001. It was analyzed by Dunn’s test using Bonferroni method. “ns” represents no significant differences.

**Figure 9 toxins-14-00879-f009:**
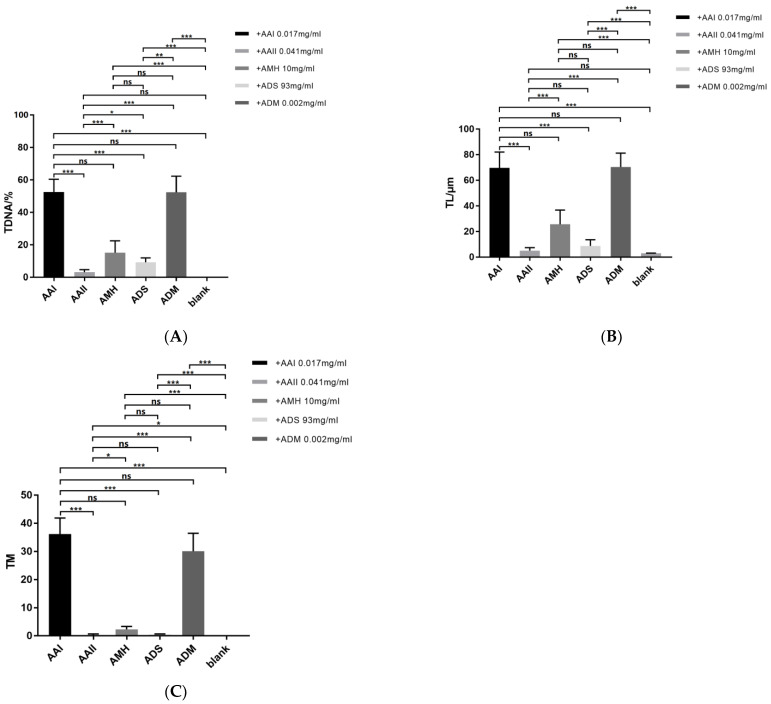
Effects of AAS/AL-S at different concentrations on DNA damage in HepG2 cells. (**A**) Comet tail DNA content; (**B**) comet tail length; (**C**) comet tail moment. Data are expressed as mean ± SD. Compared with the negative control group, * represents *p* < 0.05; ** represents *p* < 0.01; **** represents *p* < 0.0001. Statistical significance for the comet assays was determined by analysis of Kruskal–Wallis test with Bonferroni correction. “ns” represents no significant differences.

**Table 1 toxins-14-00879-t001:** Qualitative identification AA components detected in AMH and ADS in positive ion mode by UPLC-Q-TOF-MS/MS.

No	Assigned Identity	*t*_R_ (min)	Molecular Formula	Theoretical Exact Mass (Da) [M + H]^+^	Mean Measured Mass (Da) [M + H]+	Mass Accuracy (ppm)	MS/MS Fragments *^a^* (+H, +NH_4_, +Na)	Relative Content *^c^*	
								AMH	ADS
1	Aristololactamoside II	5.39	C_22_H_21_NO_9_	444.1289	444.1290	0.23	305.1373; 282.0800	0.0001	0.0101
2	Ariskanins-D	6.45	C_18_H_15_NO_7_	358.0921	358.0932	3.07	312.0938; 297.0808	-	0.0255
3	Aristolactam Ia-N-β-D-glucoside	6.56	C_22_H_19_NO_9_	442.1133	442.1129	−0.90	424.1052; 280.0604 406.1179; 250.0261	0.0977	0.0069
4	aristoloterpenate IV	6.71	C_31_H_29_NO_7_	528.2016	528.2020	0.76	550.1810 *^b^*; 474.1916; 268.1265	-	0.0235
5	10-amino-5,7-dimethoxy-aristolic II	6.72	C_18_H_15_NO_6_	342.0972	342.0966	−1.75	359.1255 *^b^*; 324.0868; 280.0918; 265.0792; 250.0534; 222.0602	0.0052	0.1829
6	aristolic acid II-8-O-β-D-glucoside	6.72	C_22_H_19_NO_12_	490.0980	490.0988	1.63	507.1232 *^a^* ;328.0457; 310.0351; 266.0458; 284.0569; 254.0581	-	0.0081
7	AL-BIII	6.93	C_18_H_15_NO_4_	310.1073	310.1081	2.58	327.1300 *^a^*; 295.0984; 267.0758; 240.1402	0.0003	0.0229
8	Aristolactam-CIV	7.28	C_18_H_15_NO_5_	326.1023	326.1020	−0.92	343.1288 *^a^*; 311.1000; 296.1593; 268.1546	0.0001	0.0203
9	aristoliukine A	7.42	C_17_H_13_NO_5_	312.0866	312.0856	−3.20	329.1118 *^a^*; 297.0688; 269.1190; 239.1677	0.0680	0.0529
10	Aristolochic acid A methyl ester	7.52	C_18_H_14_O_5_	311.0914	311.0918	1.29	328.1188 *^a^*; 295.0831; 265.0746	0.0003	0.0124
11	aristolic acid I	7.53	C_17_H_12_O_5_	297.0758	297.0747	−3.70	279.1701; 251.1017	0.0001	0.0062
12	ariskanin E	7.59	C_19_H_17_NO_7_	372.1077	372.1069	−2.15	389.1342 *^a^*; 357.0963; 341.1408; 295.1065	0.0015	-
13	Aristolochic acid IVa-6-O-β-glucoside	7.64	C_23_H_21_NO_13_	520.1085	520.1070	−2.88	358.0526; 340.0421; 312.0598; 297.0611; 296.0503	0.0127	0.0369
14	Aristolochic acid IVb	7.66	C_17_H_11_NO_8_	358.0557	358.0553	−1.12	340.0449; 312.0627; 297.0418; 296.0634	-	0.0155
15	sauristolactam	7.66	C_17_H_13_NO_3_	280.0968	280.0972	1.43	297.1232; 265.0949; 237.0810	0.0036	0.0103
16	aristchamic B	7.71	C_16_H_17_NO_4_	288.1230	288.1225	−1.74	305.1509 *^a^*; 273.1294; 258.0820	-	0.0061
17	Aristolactam A IIIa	7.73	C_16_H_11_NO_4_	282.0760	282.0765	1.77	267.0487; 239.0537	0.0750	0.0206
18	aristololide	7.84	C_17_H_10_O_5_	295.0601	295.0598	−1.02	312.0858 *^a^*; 280.0907; 262.1082	0.0001	0.0055
19	Ariskanins-E	7.95	C_19_H_17_NO_7_	372.1077	372.1064	−3.49	389.1345 *^a^*; 356.1060; 310.0765; 295.0822	0.0012	0.0119
20	Aristolactam IIIa-N-β-D-glucoside	8.17	C_22_H_19_NO_9_	442.1133	442.1128	−1.13	424.1003; 280.0552; 406.0905	0.2285	0.0864
21	Aristolactam Ia	8.80	C_16_H_9_NO_4_	280.0604	280.0604	0.00	250.0352; 222.0387	0.1126	0.0213
22	Aristolactam II-N-β-D-glucoside	9.11	C_22_H_19_NO_8_	426.1183	426.1166	−3.99	264.0580; 408.1058; 276.0613; 390.0950	0.3187	0.0215
23	Aristolochic acid III	9.37	C_17_H_11_NO_7_	342.0608	342.0609	0.29	324.0474; 280.0617; 265.0494; 298.0739;	-	1.9833
24	Aristolactam I-N-β-D-glucoside	9.49	C_23_H_21_NO_9_	456.1289	456.1290	0.22	438.1190; 420.1052; 336.0734; 294.0594	0.9799	0.2481
25	6-methoxydenitroaristolochic acid methyl ester	9.59	C_19_H_16_O_6_	341.1019	341.1031	3.52	323.1307; 308.1044; 264.1317; 249.1511; 234.1575	0.0005	0.0066
26	AAC	10.21	C_16_H_9_NO_7_	328.0452	328.0452	0.00	345.0703 *^a^*; 350.0271 *^b^*; 310.0384; 284.0603; 282.0522; 266.0446	0.0398	0.0035
27	aristofolin E	10.29	C_17_H_12_O_4_	281.0808	281.0819	3.91	263.0935; 235.0994; 251.1055	0.0011	-
28	Aristolactam III	10.30	C_17_H_11_NO_4_	294.0761	294.0767	2.04	279.0579; 251.0633	0.1084	0.0264
29	Ariskanin-B	10.77	C_17_H_13_NO_7_	344.0764	344.0761	−0.87	366.0602 *^b^*; 298.0809; 283.0614; 268.0721	-	0.0571
30	Aristolactam IIIa	10.81	C_16_H_9_NO_4_	280.0604	280.0600	−1.43	250.0198; 222.0217	0.0001	0.0065
31	AAD	11.55	C_17_H_11_NO_8_	358.0557	358.0562	1.40	375.0838 *^a^*; 380.0384 *^b^*; 340.0485; 312.0669; 296.0948; 314.0676	0.0742	0.1908
32	7-hydroxyl-8-methoxyaristolate	11.66	C_17_H_12_O_6_	313.0706	313.0716	3.19	295.0617; 280.0625; 269.0857; 241.0516	0.0003	0.0315
33	Aristolactam AII	11.74	C_16_H_11_NO_3_	266.0811	266.0812	0.38	251.0534; 223.0597; 195.0631; 167.0700	0.0007	-
34	9-OH-aristolochic acid I	11.84	C_17_H_11_NO_8_	358.0557	358.0561	1.12	340.0422; 312.0598; 297.0386	0.1116	-
35	aristolactam-CV	11.84	C_17_H_13_NO_4_	296.0917	296.0932	5.07	281.0697; 253.0506; 223.0640	0.0001	-
36	Aristolochic acid E	12.12	C_17_H_11_NO_8_	358.0557	358.0561	1.12	340.0435; 296.0542; 312.0605; 297.0392	0.2942	-
37	AL-FI	12.14	C_16_H_11_NO_3_	266.0811	266.0814	1.13	251.0567; 195.0679; 167.0712	0.0138	-
38	9-ethoxy-7-methoxy-aristololactam IV	12.56	C_21_H_19_NO_7_	398.1234	398.1251	4.27	383.1520; 368.1853	-	0.0486
39	Aristolochic acid II methyl ester	12.69	C_17_H_11_NO_6_	326.0659	326.0663	1.23	280.0671; 308.0542; 293.0310	0.0012	0.0072
40	Aristolactam IVa	13.54	C_17_H_11_NO_5_	310.0710	310.0698	−3.87	327.0986 *^a^*; 295.0488; 267.0545	0.0658	0.0290
41	Cepharanone-C	13.98	C_17_H_11_NO_5_	310.0710	310.0699	−3.55	327.0980 *^a^*; 295.0586; 267.0719	0.0001	-
42	Aristolactam IV	14.84	C_18_H_13_NO_5_	324.0866	324.0869	0.93	294.0402; 279.0556; 264.0662	0.0001	0.3941
43	Aristolactam II	14.94	C_16_H_9_NO_3_	264.0655	264.0668	4.92	234.0726; 206.0454; 179.0507	0.0651	0.0205
44	AL-BII	15.67	C_17_H_13_NO_3_	280.0968	280.0961	−2.50	302.0732 *^b^*; 265.0693; 222.0541	0.0001	-
45	AL-I	15.76	C_17_H_11_NO_4_	294.0761	294.0774	4.42	279.0579; 251.0633; 221.0560	0.0176	0.0324
46	AAII	15.78	C_16_H_9_NO_6_	312.0503	312.0518	4.81	329.0840 *^a^*; 334.0386*^b^*; 294.0435; 266.0635; 250.0540	0.2654	0.0258
47	AAI	16.43	C_17_H_11_NO_7_	342.0608	342.0623	4.39	359.0888 *^a^*; 364.0431 *^b^*; 324.0530; 296.0707; 298.0743; 280.0626; 265.0397	1.0000	0.0257
48	AAVII/AAIV	16.66	C_18_H_13_NO_8_	372.0713	372.0709	−1.08	389.1003 *^a^*; 326.0825; 354.0637; 311.0778; 283.0682	0.2120	0.0052
49	2-hydroxy-8-methoxycepharanone-A	16.77	C_17_H_11_NO_6_	326.0659	326.0664	1.53	310.0591; 282.0711; 252.0639	0.1977	-
50	3-hydroxy-4-methoxy-10-nitrophenanthrene-1-carboxylic acid methyl ester	16.79	C_17_H_13_NO_6_	328.0815	328.0830	4.57	310.0716; 295.0525; 251.0613	0.0001	-
51	aristolide C	17.11	C_17_H_11_NO_5_	310.0710	310.0706	−1.29	327.0868 *^a^*; 309.0756; 294.0767	0.0027	0.0205
52	Enterocarpam III	17.49	C_18_H_13_NO_6_	340.0815	340.0825	2.94	310.0713; 295.0800	0.0001	0.0015

Note: *^a^* Represents the ion fragment of [M + NH_4_]^+^; *^b^* Represents the ion fragment of [M + Na]^+^; *^c^* The average peak area of AAI in 16 batches of AMH is regarded as 1, so the relative content means the ratio of the average peak area of other AAs to the average peak area of AAI.

**Table 2 toxins-14-00879-t002:** Precision, stability, repeatability and recovery of 7 AAs standards.

Constituent Name	Precision (RSD%, *n* = 6)	Stability (RSD%, *n* = 6)	Repeatability (RSD%, *n* = 6)	Recovery (*n* = 6)	
				%	RSD%
AA I	1.95	1.76	2.04	99.43	3.91
AA II	1.81	3.16	1.14	101.79	3.87
AA C	3.29	3.62	3.21	100.27	3.98
AA D	1.56	2.89	3.54	97.23	4.62
AL-I	1.83	3.77	0.98	102.12	4.17
AL-BII	4.13	3.49	3.81	102.26	4.68
AL-FI	0.83	3.77	0.82	103.19	4.57

**Table 3 toxins-14-00879-t003:** Linear relationship of 7 AAs in AMH and ADS.

Peak Number	Analytes	Calibration Curves	Working Range (μg/mL)	*R* ^2^	LOD (μg/mL)	LOQ (μg/mL)
1	AAI	Y = 0.0862 X + 0.0657	1.260–126.000	0.9995	0.060	0.080
2	AAII	Y = 0.1735 X + 0.0347	0.810–81.000	0.9996	0.600	0.700
3	AAC	Y = 0.184 6 X − 0.0235	0.180–18.000	0.9989	0.120	0.150
4	AAD	Y = 0.6241 X − 0.0592	0.270–27.000	0.9992	0.080	0.110
5	AL-I	Y = 11.825 X − 0.1222	0.090–9.000	0.9994	0.004	0.005
6	AL-BII	Y = 49.495 X − 0.0262	0.002–0.180	0.9991	0.001	0.002
7	AL-FI	Y = 16.987 X − 0.5623	0.081–8.100	0.9995	0.050	0.060

**Table 4 toxins-14-00879-t004:** IC_50_ values (µM) of the cytotoxic activity of AAs compounds and the extracts.

IC_50_ Values of Compounds (µM) and the Extracts (µg/mL) (*n* = 3)					
Compounds	AAI	AAII	AAC	AAD	AL-I
IC_50_ values	50.2 ± 0.2	131.4 ± 0.1	293.5 ± 0.3	---	135.7 ± 0.3
Compounds and extracts	AL-BII	AL-FI	Extract of AMH	Extract of ADS	Extract of ADM
IC_50_ values	0.2 ± 0.1	417.2 ± 0.2	9.7 ± 0.3	92.9 ± 0.3	3.1 ± 0.1

Adriamycin and DMSO were used as positive and negative controls, respectively, and the data are expressed as means ± SD (*n* = 3). “---” indicates IC_50_ values more than 1 mM.

**Table 5 toxins-14-00879-t005:** Mass spectrometry parameters of each component.

Constituent Name	Parent Ion *m*/*z*	Daughter Ion *m*/*z*	Collision Energy eV
AAΙ	342.06	296.09	10.00
AAΙΙ	312.05	266.15	15.00
AA C	328.04	282.08	15.00
AA D	358.05	312.19	15.00
AL-Ι	294.07	279.12	30.00
AL-FΙ	266.08	251.07	20.00
AL-BΙΙ	280.09	264.09	25.00
Indomethacin	358.08	174.08	10.00

## Data Availability

Not applicable.

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
