# Peer review of "Comparative Analysis of Aristolochic Acids in Aristolochia Medicinal Herbs and Evaluation of Their Toxicities"

_toxins, 2022, doi:10.3390/toxins14120879_

Round 1
Reviewer 1 Report
The submitted manuscript aimed to develop a workflow to identify and quantitate aristolochic acids (AAs) and aristolactams (ALs) in Aristolochia medicinal herbs, as well as investigating the cytotoxicity and genotoxicity of AAs and ALs using human cell model. The objective of this study is clear, and the experimental design looks appropriate. However, the manuscript was not prepared carefully and conclusion from this study is not convincing. Current from of the manuscript cannot be considered for publication in Toxins.
Specific comments:
1. There are a lot of grammatical mistakes, typos, and missing information in the manuscript. E.g., L25: "Aristolochic lactam" should be "aristolactam"; L37: "Aristolotaceae" should be "Aristolochiaceae"; L93: "resolustion" should be "resolution"; L110: "Astrolactam" should be "Aristolactam"; L312: "at concentration of"?. I suggest the authors to check the manuscript carefully all over again.
2. Figure 1, hydrogen is missing the chemical structures for AL BII, AA FI, and AL I. There should be a hydrogen attached to the nitrogen of the lactam ring.
3. L51: AAs are classified as Group I carcinogens but not class I.
4. L53: "with abundant contents" of what?
5. Table 1: It will be better if you can designate the type of adducts for the fragments. Also, please adjust the format of the table to separate the fragment ions of different compounds. The current view is a bit confusing. What is the unit for the relative content? What does the asterisk stand for in the table?
6. The contents of AAs and ALs are greatly affected by the microenvironment of the cultivation fields. There could be difference in the composition profile of AAs and ALs in medicinal plants cultivated from the same field. I am not sure how effective the chemometric results can help distinguish types of plants and their origin. Did you perform a blind test using this workflow?
7. Section 4.2: The procedure should not be written using imperative sentences.
8. Section 4.3: L402-403: Why is there a detection wavelength for QTOF? L409: redundant.
9. Table 4: How do you calculate molarity of extract of AMH, ADS, and ADM?
10. Figure 8: What do you mean by quantitative AAs?
11. What are the advantages for your developed method compared to other published methods in the literature? I cannot see any novelty in your method.
Author Response
Manuscript Number: Toxins-2052805
TITLE: Comparative analysis of aristolochic acids in Aristolochia medicinal herbs and evaluation of their toxicities
Dear Editor and Reviewers:
We thank the reviewers for their insightful comments. We have rewritten and addressed the reviewer’s issues thoroughly with yellow highlights and blue font in the revised edition. We have polished the manuscript by the language editing service. We believe that the changes made according to your suggestions have significantly strengthened the manuscript.
Response to the comments of Reviewer #1
Comment 1: There are a lot of grammatical mistakes, typos, and missing information in the manuscript. E.g., L25: "Aristolochic lactam" should be "aristolactam"; L37: "Aristolotaceae" should be "Aristolochiaceae"; L93: "resolustion" should be "resolution"; L110: "Astrolactam" should be "Aristolactam"; L312: "at concentration of"?. I suggest the authors to check the manuscript carefully all over again.
Response: Thank you for your comment. We have revised the mistakes that you mentioned. Moreover, we’ve checked the spelling mistakes throughout the whole manuscript and improved the manuscript’s English by the editing service.
Comment 2: Figure 1, hydrogen is missing the chemical structures for AL BII, AA FI, and AL I. There should be a hydrogen attached to the nitrogen of the lactam ring.
Response: We have supplemented the hydrogen in Figure 1.
Comment No. 3: L51: AAs are classified as Group I carcinogens but not class I.
Response: We have revised it.
Comment No. 4: L53: "with abundant contents" of what?
Response: We have corrected the phrase as follows. “Although AAs-containing herbs with abundant varieties, …”
Comment No. 5: Table 1: It will be better if you can designate the type of adducts for the fragments. Also, please adjust the format of the table to separate the fragment ions of different compounds. The current view is a bit confusing. What is the unit for the relative content? What does the asterisk stand for in the table?
Response: We have removed the column of the adducts and added the +NH4 and +Na adducts behind the corresponding MS/MS ion fragments in Table 1. Additionally, we supplemented the note “c” of “the relative content” below Table 1, as well we removed the asterisks. Please check in the manuscript.
Comment No. 6: The contents of AAs and ALs are greatly affected by the microenvironment of the cultivation fields. There could be difference in the composition profile of AAs and ALs in medicinal plants cultivated from the same field. I am not sure how effective the chemometric results can help distinguish types of plants and their origin. Did you perform a blind test using this workflow?
Response: We agree with your opinion that the microenvironment may affect the contents of AAs and ALs, which is the reason why we prefer to select the herbs with good quality from the genuine producing areas. In this paper, we collected the three species herbs from their genuine producing areas in order to evaluate the AAs content objectively. Moreover, we also acquired the ideal data, which the multivariate data analysis (PCA and OPLS-DA) of simultaneous determination of seven AAs components indicated a clear separation of samples from different origins. PCA is computed without regard to any underlying structures influenced by the latent variables. In addition, components are calculated using all the variance of the manifest variables in PCA, so that the whole variance appears in the solution. Thus, the results of principal component analysis as s blind test are reliable.
Comment No. 7: Section 4.2: The procedure should not be written using imperative sentences.
Response: We have re-written this section according to your comments. Please check the manuscript.
Comment No. 8: Section 4.3: L402-403: Why is there a detection wavelength for QTOF? L409: redundant.
Response: We have already deleted the detection wavelength in Line 402 and the redundant sentence in Line 409.
Comment No. 9: Table 4: How do you calculate molarity of extract of AMH, ADS, and ADM?
Response: We have revised the molarity of the extracts using μg/mL.
Comment No. 10: Figure 8: What do you mean by quantitative AAs?
Response: We have revised the sentence of Figure 8. Please check the sentence as follows.
Figure 8. Content differences of the AAs in AMH, ADS and ACY from different origins (8A- 8C), respectively; and content differences in 3 species of medicinal herbs (8D). A. AMH of Hubei and Shandong provinces; B. ADS of Guangxi and Hubei provinces; C. ACY of Sichuan and Yunnan provinces; D. 3 medicinal herbs of AMH, ADS and ACY. Data are expressed as the mean ± SD. *: p<0.05 and **: p<0.01 analyzed by Mann Whitney U test.#: p<0.05 and ###: p<0.001 analyzed by Dunn’s test using Bonferroni method. “ns” represents no significant differences.
Comment No. 11: What are the advantages for your developed method compared to other published methods in the literature? I cannot see any novelty in your method.
Response: In this study, we used the widely non-targeted metabolomics method for the direct chemical screening of AAs/ALs in 3 species of Aristolochia using LC-MS/MS analysis. Our analysis showed that there were 44, 41, and 45 kinds of AAs, respectively, belonging to AMH, ADS and ACY. Furthermore, we found that the contents of 26 AAs and ALs in ACY were higher than those in ADS and AMH. Multivariate data analysis (PCA and OPLS-DA) indicated a clear separation of the samples according to different species and origins, indicating the results obtained in the simultaneous determination of seven AAs components. Therefore, the untargeted metabolomics approach was particularly useful for the distinguish three different species and origins.

Reviewer 2 Report
Toxins #2052805
In this manuscript, authors aim to develop quantitative methods for evaluation of the content of aristolochic acids (AAs) and aristolactams (ALs) in several Aristolochia medicinal herbal remedies, such as A. molissima (AMH), A. debilis (ADS) and A. cinnabaria (ACY). Furthermore, cytotoxicities and genotoxicities of seven AAs and ALs standards, AMH and ADS extracts were evaluated using human hepatic HepG2 cells.
This research is highly significant because certain AAs and ALs represent health hazard as carcinogenic, cytotoxic, hepatotoxic and nephrotoxic compounds, and numerous species of Aristolochia plants have been widely used in Chinese Herbal Remedies and in some Chinese patent drugs. However, there are no robust methods for the rapid, accurate and simultaneous evaluation of AAs and ALs content in medicinal preparations of Aristolochia. Also, no systematic knowledge of AAs and ALs species across herbal preparations of Aristolochia has been established. Development of such approaches and knowledge will greatly contribute to safety testing of medicinal herbs.
In this work, authors report on the UPLC-QTOF-MS/MS technique that allows to determine qualitatively multiple AAs and ALs species in AMD, ADS and ACY herbal preparations, each preparation represented by over a dozen of batches originating from two different provinces; Hubei and Shandong for AMH, Hubei and Guangxi for ADS and Sichuan and Yunnan for ACY. Interestingly, some of these ALs and AAs species were unique for certain herbal preparations, also, permitting stratification by province origin. Importantly, AMH showed higher cytotoxicity and genotoxicity than ADS, which is likely due to the higher AA-I content in the AMH sample as compared to ADS. Therefore, authors conclude that AMH remedies have to be better controlled in terms of safety.
While this work is important and has significant value for public health, the manuscript requires significant improvements in terms of language and description of experimental work and results. Below provided detailed comments regarding these issues.
There are numerous typographical errors in the manuscript. Take one example, Fig 1 legend: “aristolotacea”. The correct way is likely “aristolochiacea”.
Materials and Methods section in some parts is written as a research protocol (e.g. “draw a standard curve…” instead of “a standard curve was drawn”).
Some figure legends could be improved to provide more information on experimental design. For example: Figure 2, are spectra presented for standard compounds?
In many cases description of results do not refer to particular segments of their respective figures. For example, 2.2.2 Results of PCA and OPLS-DA: This section needs significant improvement. Description of Figure 5 is somewhat uncertain (lines 167-177). It would be helpful if the particular parts of figures 4 and 5 (A, B, C or D) were indicated in the text following description of corresponding results.
Figure 6: Red bars are VIP >1. What are the green bars?
Table 4: How was uM value defined for AMH and ADS? Which batches from which provinces were used for these assays? Is genetoxicity or cytotoxicity different between provinces and batches? What are the ACY values for the MTT and Comet assay? Authors mention in conclusions about the genotoxicity of ACY due to its high AA-I content but do not show these data in results. In general, some data for ACY are presented in some cases and in some do not, which is somewhat confusing.
Do authors know why different provinces differ in the AAs/ALs content? Is that because methods how these herbs prepared differ between the regions, or it is a climate variation leading to a different AAs/ALs content?
Figure 8: Authors conclude that the origins have obvious impact on the AAs/ALs content of the three herbal preparations. However, from the presented graphs for these particular compounds it seems like it could be the case for AMH, but not ADH and ACY. Thus for ADS (Fig 8B), AA-I and AA-II look similar for Guangxi and Hubei and only AAD and to a lesser extent AL-I look like showing the difference. For ACY (Fig 8C), all analyzed AAs and AL-I appears to be similar between the two origins. Perhaps, this paragraph needs to be revised to better reflect conclusions of these findings.
Author Response
Manuscript Number: Toxins-2052805
TITLE: Comparative analysis of aristolochic acids in Aristolochia medicinal herbs and evaluation of their toxicities
Dear Editor and Reviewers:
We thank the reviewers for their insightful comments. We have rewritten and addressed the reviewer’s issues thoroughly with yellow highlights and blue font in the revised edition. We have polished the manuscript by the language editing service. We believe that the changes made according to your suggestions have significantly strengthened the manuscript.
Response to the comments of Reviewer #2
Comment No.1: There are numerous typographical errors in the manuscript. Take one example, Fig 1 legend: “aristolotacea”. The correct way is likely “aristolochiacea”
Response: Thank you for your comment. We have revised the mistakes that you mentioned. Moreover, we’ve checked the spelling mistakes throughout the whole manuscript and improved the manuscript’s English by the editing service.
Comment No.2: Materials and Methods section in some parts is written as a research protocol (e.g. “draw a standard curve…” instead of “a standard curve was drawn”).
Response: We have revised the corresponding part in the section of “Materials and Methods”. Please check the manuscript.
Comment No.3: Some figure legends could be improved to provide more information on experimental design. For example: Figure 2, are spectra presented for standard compounds?
Response: We have supplemented more details for the figure legends to illustrate the meaning of each Figure. Please check the legends below the Figures from 2 to 8 in the manuscript.
Comment No.4: In many cases description of results do not refer to particular segments of their respective figures. For example, 2.2.2 Results of PCA and OPLS-DA: This section needs significant improvement. Description of Figure 5 is somewhat uncertain (lines 167-177). It would be helpful if the particular parts of figures 4 and 5 (A, B, C or D) were indicated in the text following description of corresponding results.
Response: We have added more description and details in the results of PCA and OPLS-DA (2.2.2). You can also check the description from Line 161 to Line 189 in the manuscript. Moreover, the explanation of Figure 5 has been supplemented in both the Figures 4 and 5 and the section 2.2.2.
Comment No. 5: Figure 6: Red bars are VIP >1. What are the green bars?
Response: We have revised the Figure legend as follows. “AAs with the variable importance in projection (VIP) scores greater than 1 were considered as the important compounds towards the classification model and were marked in red. Compounds with VIP<1 were marked in green.”
Comment No. 6: Table 4: How was uM value defined for AMH and ADS? Which batches from which provinces were used for these assays? Is genetoxicity or cytotoxicity different between provinces and batches? What are the ACY values for the MTT and Comet assay? Authors mention in conclusions about the genotoxicity of ACY due to its high AA-I content but do not show these data in results. In general, some data for ACY are presented in some cases and in some do not, which is somewhat confusing.
Response: We have revised the molarity of the AMH and ADS extracts using μg/mL. As to the cytotoxicity and genetoxicity assays, we selected two batches for each herb that contained the lowest content of AAâ… ( batches of 20211121 and 20200801 for AMH and ADS, respectively). We have analyzed using LC-MS that ACY contains the highest content of AAâ… and the highest content of AAs among the three medicinal herbs, which it is the most toxic herb. In addition, ACY (Aristolochia cinnabarina) has been forbidden in China, so we didn’t evaluate its toxicity for further study. To elucidate the genetoxicity of the AAs and the extracts of AMH and ADS more specifically, we used statistical method to analyze for the results of the comet assay via Kruskal-Wallis test with Bonferroni correction. Please check the manuscript.
Comment No. 7: Do authors know why different provinces differ in the AAs/ALs content? Is that because methods how these herbs prepared differ between the regions, or it is a climate variation leading to a different AAs/ALs content?
Response: People usually prefer to select the herbs with good quality from the genuine producing areas, which means geographical environment may influence the quality of the medicinal herbs. Since the preparation methods of all samples are the same in this study, the differences in the determination results may be due to the geographical environment and climate. The results of our study showed that the content and composition of AAs of the Aristolochia herbs are obviously different from the different geographical areas.
Comment No. 8: Figure 8: Authors conclude that the origins have obvious impact on the AAs/ALs content of the three herbal preparations. However, from the presented graphs for these particular compounds it seems like it could be the case for AMH, but not ADH and ACY. Thus for ADS (Fig 8B), AA-I and AA-II look similar for Guangxi and Hubei and only AAD and to a lesser extent AL-I look like showing the difference. For ACY (Fig 8C), all analyzed AAs and AL-I appears to be similar between the two origins. Perhaps, this paragraph needs to be revised to better reflect conclusions of these findings.
Response: We have revised this part according to the your comments. We performed statistical analysis on the content of AAS from different producing areas. The normality test results showed that the samples from AMH, ADS and ACY do not meet the normality test (P<0.05), so it is recommended to select Mann Whitney U test. The contents of AAI, AAII, AAC and AL-I of AMH from Shan Dong were significantly decreased compared to the Hu Bei group, while the contents of AAD, AL-BII and AL-FI showed no differences between the two origins.
The results of Mann Whitney U test revealed significant differences in the content of AAC, AAD and AL-I of ADS between the Hubei and Guangxi, while the contents of AAI and AAII showed no differences. The contents of AAD and AL-I of ADS in Hubei group significantly decreased as compared to Guangxi group (p <0.001), the content of AAC in Hubei group significantly increased (p <0.05). The results of Mann Whitney U test showed no differences in the content of AAI, AAC, AAD and AL-I of ACY between the Yunan and SiChuan while the content of AAII in Yunan was remarkably higher than that of the Sichuan (p <0.05).
We also compared the contents of AAs/ALs from the three medicinal materials. The normality test results showed that the samples from AMH, ADS and ACY do not meet the normality test (P<0.05), so it is recommended to select Kruskal-Wallis Test. The results showed that there was a significant difference in the contents of AAs/Als in three kinds of medicinal materials. Afterwards, we performed the Dunn’s test using Bonferroni method to correct the significance. The results showed that the contents of AAs/Als were significantly highest in ACY and lowest in ADS besides the contents of AAD and AL-I in AMH. The content of AAD in ADS was remarkably higher than that in AMH, while there was no difference in the content of AL-I.
